# Self-reported sleep duration and napping, cardiac risk factors and markers of subclinical vascular disease: cross-sectional study in older men

Shahrzad Zonoozi,[1] Sheena E Ramsay,[2] Olia Papacosta,[1] Lucy Lennon,[1] Elizabeth A Ellins,[3] Julian P J Halcox,[3] Peter H Whincup,[4] S Goya Wannamethee[1]

[1]UCL Department of Primary Care and Population Health, UCL Medical School, London, UK
[2]Institute of Health & Society, Newcastle University, Newcastle upon Tyne, UK
[3]Institute of Life Sciences, Swansea University, Swansea, UK
[4]Population Health Research Institute, St George's University of London, London, UK

**Correspondence to**
Dr Shahrzad Zonoozi;
shahrzadz@gmail.com

## ABSTRACT

**Study objectives** Daytime sleep has been associated with increased risk of cardiovascular disease and heart failure (HF), but the mechanisms remain unclear. We have investigated the association between daytime and night-time sleep patterns and cardiovascular risk markers in older adults including cardiac markers and subclinical markers of atherosclerosis (arterial stiffness and carotid intima-media thickness (CIMT)).

**Methods** Cross-sectional study of 1722 surviving men aged 71–92 examined in 2010–2012 across 24 British towns from a prospective study initiated in 1978–1980. Participants completed a questionnaire and were invited for a physical examination. Men with a history of heart attack or HF (n=251) were excluded from the analysis.

**Results** Self-reported daytime sleep duration was associated with higher fasting glucose and insulin levels (p=0.02 and p=0.01, respectively) even after adjustment for age, body mass index, physical activity and social class. Compared with those with no daytime sleep, men with daytime sleep >1 hour, defined as excessive daytime sleepiness (EDS), had a higher risk of raised N-terminal pro-brain natriuretic peptide of ≥400 pg/mL, the diagnostic threshold for HF (OR (95% CI)=1.88 (1.15 to 3.1)), higher mean troponin, reduced lung function (forced expiratory volume in 1 s) and elevated von Willebrand factor, a marker of endothelial dysfunction. However, EDS was unrelated to CIMT and arterial stiffness. By contrast, night-time sleep was only associated with HbA1c (short or long sleep) and arterial stiffness (short sleep).

**Conclusions** Daytime sleep duration of >1 hour may be an early indicator of HF.

## INTRODUCTION

Sleep disturbance is a common complaint, especially among older adults.[1] It is increasingly recognised that there is a U-shaped association between sleep duration and risk of cardiovascular disease (CVD) and metabolic disturbances, that is both short and long sleep durations are associated with an increased risk.[2–4] More recently, studies have shown excessive daytime sleepiness (EDS), one of the more common sleep disturbances

---

**Strengths and limitations of this study**

► To the best of our knowledge, this is the first study to look at the association between excessive daytime sleepiness, cardiac markers and vascular parameters.
► Owing to a lack of data, we do not know if the associations seen are true for women, middle-aged populations and ethnic minorities.
► As we do not have polysomnography data, we were unable to fully exclude obstructive sleep apnoea as a contributing factor to our findings.
► Our results are based on a cross-sectional analysis and therefore causality cannot be established.

reported in the older population, to be associated with increased risks of CVD,[5 6] heart failure (HF)[6 7] and all-cause mortality.[8–10]

The mechanism of the association between EDS and CVD risk remains unclear. While some studies have looked at the association between EDS and metabolic risk markers,[11 12] few have examined mechanisms by which EDS may influence CVD and in particular incident HF.[6] In particular, the association between EDS and markers of arterial disease and surrogate markers of cardiac function, including N-terminal pro-brain natriuretic peptide (NT-pro-BNP), a marker of ventricular stress, and cardiac troponin T (cTnT), a marker of myocardial injury, have been less studied. Subclinical arterial disease markers are increasingly being used as early predictors for atherosclerosis and CVD. A number of non-invasive assessments have been developed, including measures of atherosclerosis such as carotid intima-media thickness (CIMT) and indirect measures of arterial stiffness including carotid distensibility and pulse wave velocity (PWV). Although a number of studies have examined the association between night-time sleep

duration and arterial stiffness,[13–17] studies on the association between EDS and vascular risk markers are limited.[5] We have therefore examined the association between daytime sleep as well as night-time sleep with vascular risk factors including metabolic risk factors, inflammatory and endothelial markers, cardiac markers and measures of subclinical arterial disease (including arterial stiffness and CIMT).

## METHODS

The British Regional Heart Study is a prospective study that recruited a socioeconomically and geographically representative cohort of 7735 men from 24 British towns between 1978 and 1980. The present investigation is based on a reassessment in 2010–2012, when all surviving men (n=3137), aged between 71 and 92 years, were sent a postal questionnaire and invited for a 30th year re-examination. A total of 2137 (68%) men completed the postal questionnaire, and 1722 (55%) men attended the re-examination.[18] Blood samples were collected after fasting for a minimum of 6 hours and were stored at −70°C. Ethical approval for data collection was obtained from the relevant research ethics committee. The men were asked if a doctor had ever told them they had a heart attack or HF, and these men were excluded from the analysis (n=251). Of the men with no history of heart attack or HF, 1471 men had physical, blood and ultrasound measurements carried out. This present study is a cross-sectional analysis of the data from the 2010–2012 questionnaires and re-examination.

### CVD risk factors

Physical examination included blood pressure (BP), lung function as well as height and weight, from which body mass index (BMI) was calculated.[19] Height standardised forced expiratory volume in 1s (FEV1) was measured using the Vitalograph Compact II. Details of measurement and classification for smoking status and physical activity in this cohort have been previously described.[20] Details of the men's medication history, including use of antihypertensive, hypnotic and anxiolytic medications, were recorded at the examination, and participants were also asked to report symptoms of breathlessness. Measurements of metabolic, inflammatory and endothelial markers were taken as described previously.[19 21 22] NT-pro-BNP and cTnT were measured using the Elecsys 2010 electrochemiluminescence method (Roche Diagnostics, Burgess Hill, UK). The lowest detectable value of cTnT was 5 and for anyone with a recorded reading of <5, we used half of the lowest detectable value (ie, 2.5). NT-pro-BNP ≥400 pg/mL was defined as high, as per current guidelines issued by the National Institute of Clinical Excellence for the diagnosis of HF.[23] Prevalent diabetes included men with doctor-diagnosed diabetes and men with fasting blood glucose ≥7 mmol/L. A total of 1402 men had at least one blood measurement (69 missing). Chronic kidney disease (CKD) was defined as

estimated glomerular filtration rate of less than 60 ml/min/1.73m$^2$ using the CKD Epidemiology Collaboration creatinine equation.[24]

### Sleep parameters

Reported duration of night sleep was based on response to: 'On average, how many hours of sleep do you have each night?' Duration of day sleep was based on response to 'On average, how much sleep (if any) do you have during the daytime?' If a response was entered for the night sleep question but no response was indicated for the day sleep question, the answer was set to 0. A total of 1436 men completed the questions on night-time and daytime sleep duration (35 missing). Night-time sleep was categorised into three groups: <6, 6–8.9 and ≥9 hours. Daytime sleep was categorised into four groups: 0 hours, <1 hour, 1 hour and >1 hour. Those sleeping >1 hour during the daytime were defined as having EDS.

### Non-invasive markers of arterial disease

Measurements were measured by two vascular technicians in series. Left and right carotid arteries were images using a Z.One Ultra ultrasound system (Zonare Medical Systems, Mountain View, California, USA) with a 5 mHz to 10 mHz linear probe. A cross-sectional sweep from the base of the common carotid artery to the jaw bone and longitudinal images of the common carotid artery approximately 1 cm proximal to the carotid bifurcation were recorded. Peak systolic and end-diastolic common carotid artery diameter and CIMT, the distance between the leading edge of the intima and the media-adventitia interface, were measured using Carotid Analyser software (Medical Imaging Applications, Iowa City, Iowa, USA). From the longitudinal images, a region of interest (5–10 mm) was selected in a plaque free area, at least 1 cm from the bifurcation.

CIMT was measured from three end-diastolic images on each side, and a mean of these measures was calculated. Maximum and minimum carotid artery diameter was assessed from three consecutive waveforms and mean distension was calculated as: maximum diameter – minimum diameter. The distensibility coefficient was then calculated as described by Dijk et al: distensibility coefficient = [(2 × mean distension/baseline diameter)/mean pulse pressure (kPa)]*1000.[25] Carotid to femoral PWV was assessed using a Vicorder (Skidmore Medical, Bristol, UK), with participants in a semisupine position with their torso at approximately 30°. A 2×9 cm cuff was positioned around the neck with the bladder over the right carotid pulse, and a Hokanson SC10 cuff around the middle of the right thigh. Path length was measured from the sternal notch to the centre of the thigh cuff. The cuffs were simultaneously inflated and traces with a minimum of three good quality waveforms recorded. Two PWV measurements, within ≤0.5 m/s of each other, were accepted and averaged. A total of 1464 men had at least one non-invasive vascular measurement carried out (seven missing).

## Statistical analysis

Distributions of HbA1c, glucose, insulin, C reactive protein (CRP), interleukin (IL)-6, NT-pro-BNP and cTnT were highly skewed, and log transformation was used. Comparisons of baseline characteristics between the sleep groups were carried out using the $\chi^2$ test for categorical variables and analysis of variance for continuous variables. Logistic regression models were used to assess the differences in outcomes between categorical groups. In multivariate analyses, age, BMI, physical activity and social class were fitted as continuous variables, while prevalent diabetes and use of antihypertensive medications was defined as categorical variables. The base groups for the statistical analysis were the group sleeping 6–8.9 hours in the night-time sleep group. A threshold effect was seen in those sleeping >1 hour in the daytime, and therefore we compared those sleeping >1 hour with those sleeping ≤1. We also carried out statistical analysis of the trend in the daytime sleep group. All analyses were performed using SAS V.9.3.

## RESULTS

The average duration of night-time sleep in the men who responded was 6.8 hours (±1.4 hours). Fifteen per cent of the men reported night-time sleep duration of less than 6 hours, while 7% reported sleeping ≥9 hours during the night. Forty-seven per cent of men reported no daytime sleep, while 9% reported EDS. Night-time sleep duration was significantly associated with EDS (p=0.02; correlation coefficient r=−0.063), with a smaller proportion of those sleeping ≥9 hours reporting EDS. Daytime sleep duration was significantly associated with night-time sleep of less than 6 hours (p=0.02).

### Baseline characteristics

Table 1 describes the baseline characteristics of the study population according to night-time and daytime sleep patterns. A U-shaped relationship was seen between night-time sleep and CVD risk factors. Both short and long night-time sleep were associated with increased age, increased prevalence of physical inactivity, manual workers and breathlessness compared with those with 6–8.9 hours sleep. There was a significant association between night-time sleep and use of hypnotic or anxiolytic medications. Daytime sleep is strongly associated with older age and with many adverse characteristics including physical inactivity, obesity, prevalent diabetes, breathlessness, use of antihypertensive medication, depression and CKD. The prevalence of these adverse factors tended to increase with increasing duration of daytime sleep. Daytime sleep was not associated with use of hypnotic or anxiolytic medications.

### Night-time sleep and and cardiovascular risk markers

There was a U-shaped association between night-time sleep and metabolic risk factors including HbA1c and glucose (table 2). The association between night-time sleep and HbA1c remained significant following

**Table 1** Baseline characteristics of the study population a, maximum n in group, varies slightly with missing covariate data

| | | Age mean±std | Obese (%) | Cigarette smoker (%) | Inactive (%) | Manual social class (%) | Diabetic (%) | Breathlessness (%) | Using antihypertensives (%) | Depression (%) | CKD (%) |
|---|---|---|---|---|---|---|---|---|---|---|---|
| Night-time sleep duration (hours) | <6 (n=213†) | 78.76±4.6 | 21.53 | 0.94 | 48.36 | 57.35 | 18.31 | 40.72 | 53.99 | 11.43 | 29 |
| | 6–8.9 (n=1126†) | 78.22±4.6 | 18.78 | 3.83 | 33.66 | 42.4 | 14.87 | 19.32 | 48.85 | 8.39 | 29.36 |
| | ≥9 (n=97†) | 79.24±4.9 | 22.11 | 4.12 | 38.14 | 48.45 | 19.79 | 25 | 52.58 | 11.34 | 38.71 |
| | p | 0.05 | 0.52 | 0.10 | 0.0002* | 0.002* | 0.24 | <0.0001* | 0.33 | 0.27 | 0.16 |
| Daytime sleep duration (hours) | 0 (n=680†) | 77.83±4.4 | 16.1 | 2.65 | 30.74 | 46.1 | 12.24 | 17.59 | 45.88 | 7.41 | 26.42 |
| | <1 (n=290†) | 78.04±4.4 | 15.63 | 2.76 | 33.45 | 40 | 14.88 | 21.61 | 54.14 | 10.42 | 30.94 |
| | 1 (n=336†) | 78.79±4.7 | 24.17 | 5.07 | 40.77 | 44.61 | 20 | 27.01 | 49.7 | 8.98 | 32.81 |
| | >1 (n=130†) | 80.05±5 | 33.33 | 4.62 | 58.46 | 51.54 | 24.62 | 41.46 | 61.54 | 14.62 | 39.17 |
| | p | <0.0001* | <0.0001* | 0.18 | <0.0001* | 0.15 | 0.0003* | <0.0001* | 0.004* | 0.05 | 0.02* |

*denotes statistical significance.
†Maximum n in group, varies slightly with missing covariate data.
CKD, chronic kidney disease.

**Table 2** Night-time sleep duration versus metabolic risk factors, inflammatory and endothelial markers, lung function and cardiac markers

| | | Night sleep (hours) | | | |
|---|---|---|---|---|---|
| | | <6(n=213¶) | 6–8.9(n=1126¶) | ≥9(n=97¶) | p |
| **Metabolic risk factors (mean (95% CI))** | | | | | |
| SBP (mm Hg) | † | 146.04 (143.52 to 148.56) | 147.31 (146.21 to 148.4) | 150.73 (146.99 to 154.47) | 0.12 |
| DBP (mm Hg) | † | 76.59 (75.07 to 78.11) | 77.46 (76.8 to 78.12) | 77.25 (74.99 to 79.5) | 0.59 |
| HDL-C (mmol/L) | † | 1.47 (1.41 to 1.53) | 1.48 (1.45 to 1.5) | 1.37 (1.28 to 1.46) | 0.07 |
| Triglycerides (mmol/L) | † | 1.3 (1.21 to 1.39) | 1.28 (1.25 to 1.32) | 1.39 (1.26 to 1.52) | 0.31 |
| HbA1c§ (%) | † | 5.93 (5.85 to 6.02) | 5.75 (5.72 to 5.79) | 5.88 (5.76 to 6) | 0.0001* |
| | ‡ | 5.9 (5.82 to 5.98) | 5.76 (5.73 to 5.79) | 5.86 (5.74 to 5.97) | 0.003* |
| Glucose§ (mmol/L) | † | 5.78 (5.62 to 5.94) | 5.55 (5.48 to 5.61) | 5.72 (5.49 to 5.95) | 0.02* |
| | ‡ | 5.74 (5.58 to 5.9) | 5.56 (5.49 to 5.62) | 5.68 (5.45 to 5.91) | 0.08 |
| Insulin§ (mU/L) | † | 8.26 (7.54 to 9.04) | 7.92 (7.62 to 8.24) | 7.82 (6.84 to 8.94) | 0.69 |
| **Inflammatory and endothelial markers (mean (95% CI))** | | | | | |
| CRP§ (mg/L) | † | 1.51 (1.28 to 1.78) | 1.34 (1.25 to 1.44) | 1.46 (1.15 to 1.87) | 0.37 |
| IL-6§ (pg/mL) | † | 3.03 (2.74 to 3.35) | 2.98 (2.86 to 3.11) | 3.72 (3.21 to 4.3) | 0.02* |
| | ‡ | 2.94 (2.67 to 3.25) | 3 (2.87 to 3.13) | 3.61 (3.12 to 4.18) | 0.05 |
| vWF (IU/dL) | † | 140.97 (130.97 to 150.97) | 133.27 (128.91 to 137.63) | 119.04 (104.2 to 133.89) | 0.06 |
| **Lung function (mean (95% CI))** | | | | | |
| FEV1 (L) | † | 2.43 (2.35 to 2.5) | 2.46 (2.42 to 2.49) | 2.44 (2.32 to 2.55) | 0.76 |
| **Cardiac markers (mean (95% CI))** | | | | | |
| NT-pro-BNP§ (pg/mL) | † | 121.67 (101.33 to 146.1) | 126.45 (116.87 to 136.81) | 134.61 (102.8 to 176.25) | 0.83 |
| cTnT§ (pg/mL) | † | 11.25 (10.29 to 12.3) | 10.82 (10.41 to 11.24) | 10.14 (8.89 to 11.57) | 0.44 |

†Adjusted for age.
‡Adjusted for age, BMI, social class and physical activity.
§Geometeric mean.
¶Maximum n in group, varies slightly with missing covariate data.
BMI, body mass index; CRP, C reactive protein; cTnT, cardiac troponin T; DBP, diastolic blood pressure; FEV1, forced expiratory volume in 1 s; HbA1c, glycated haemoglobin; HDL-C, high density lipoprotein cholesterol; IL-6, interleukin 6; NT-pro-BNP, N-terminal pro-brain natriuretic peptide; SBP, systolic blood pressure; vWF, von Willebrand factor.

adjustment for daytime sleep (p=0.04). In age-adjusted analysis, night-time sleep was significantly associated with IL-6, a marker of inflammation, but this association was attenuated following adjustment for BMI, social class and physical activity (p=0.05). No significant association was seen between night-time sleep and von Willebrand factor (vWF), a marker of endothelial function, FEV1 (lung function) or cardiac markers (NT-pro-BNP and cTnT).

### Daytime sleep and cardiovascular risk markers
In age-adjusted analysis, metabolic risk factors including high density lipoprotein cholesterol, triglycerides, HbA1c and insulin tended to increase with increasing hours of daytime sleep (table 3). No association was seen with BP. The increasing trend was attenuated after adjustment for age, BMI, social class and physical activity. However, compared with those who slept ≤1 hour, those with EDS (>1 hour) showed significantly higher mean levels of HbA1c, glucose and insulin (table 3). Mean levels of glucose and insulin remained significantly higher in those with EDS (p=0.03 and 0.01, respectively) even after

further adjustment for prevalent diabetes and night-time sleep.

CRP increased with increasing daytime sleep duration, even after full adjustment (p trend=0.04). Lung function decreased with increasing daytime sleep duration with levels markedly reduced in those with EDS compared with those reporting ≤1 hdaytime sleep (p<0.0001). Only EDS was associated with significantly higher mean vWF levels (p=0.03). When we examined the odds of having high vWF (top quintile) and low FEV1 (lowest quintile), those with EDS were 1.8 times more likely to have a high vWF (OR (95% CI)=1.82 (1.17 to 2.83)) and 2.4 times more likely to have a low FEV1 (OR (95% CI)=2.4 (1.53, 3.74)) compared with those who reported no daytime sleep.

Mean levels of NT-pro-BNP and cTnT (cardiac markers) were significantly higher only in those with EDS following full adjustment (p=0.02 and p=0.03 respectively). This association remained significant even after adjustment for CKD. Men with EDS were more likely to have high (top quintile) mean NT-pro-BNP (OR (95% CI)=1.69

**Table 3** Daytime sleep duration versus metabolic risk factors, inflammatory and endothelial markers, lung function and cardiac markers

| | | Daytime sleep duration (hours) | | | | p Value trend | p Value >1hour versus others |
|---|---|---|---|---|---|---|---|
| | | 0(n=680¶) | <1(n=290¶) | 1(n=336¶) | >1(n=130¶) | | |
| **Metabolic risk factors (mean (95% CI))** | | | | | | | |
| SBP (mm Hg) | † | 147.46 (146.05 to 148.88) | 148.76 (146.59 to 150.92) | 147.14 (145.13 to 149.15) | 144.13 (140.88 to 147.39) | 0.90 | 0.04* |
| | ‡ | 147.49 (146.07 to 148.9) | 148.59 (146.42 to 150.75) | 146.97 (144.95 to 148.99) | 144.78 (141.44 to 148.11) | 0.96 | 0.12 |
| DBP (mm Hg) | † | 77.41 (76.56 to 78.26) | 78.08 (76.77 to 79.38) | 77.16 (75.95 to 78.37) | 75.54 (73.57 to 77.5) | 0.98 | 0.06 |
| | ‡ | 77.56 (76.7 to 78.41) | 78.04 (76.74 to 79.35) | 76.94 (75.73 to 78.16) | 75.64 (73.63 to 77.65) | 0.74 | 0.09 |
| HDL-C (mmol/L) | † | 1.5 (1.47 to 1.54) | 1.47 (1.42 to 1.52) | 1.44 (1.39 to 1.49) | 1.33 (1.25 to 1.41) | 0.01* | 0.0002* |
| | ‡ | 1.49 (1.46 to 1.52) | 1.46 (1.41 to 1.5) | 1.46 (1.42 to 1.51) | 1.41 (1.33 to 1.48) | 0.14 | 0.10 |
| Triglycerides (mmol/L) | † | 1.24 (1.19 to 1.29) | 1.3 (1.23 to 1.38) | 1.35 (1.28 to 1.42) | 1.42 (1.31 to 1.54) | 0.01* | 0.02* |
| | ‡ | 1.26 (1.21 to 1.3) | 1.33 (1.26 to 1.4) | 1.33 (1.26 to 1.4) | 1.34 (1.22 to 1.45) | 0.03* | 0.47 |
| HbA1c (%) | † | 5.73 (5.69 to 5.78) | 5.74 (5.68 to 5.81) | 5.87 (5.8 to 5.93) | 5.99 (5.88 to 6.1) | 0.01* | 0.0003* |
| | ‡ | 5.75 (5.7 to 5.79) | 5.77 (5.7 to 5.83) | 5.84 (5.78 to 5.91) | 5.9 (5.79 to 6.01) | 0.04* | 0.04* |
| Glucose§ (mmol/L) | † | 5.52 (5.43 to 5.6) | 5.48 (5.36 to 5.61) | 5.7 (5.58 to 5.83) | 5.98 (5.77 to 6.2) | 0.09 | 0.0002* |
| | ‡ | 5.54 (5.45 to 5.62) | 5.51 (5.38 to 5.63) | 5.68 (5.56 to 5.8) | 5.86 (5.65 to 6.07) | 0.25 | 0.01* |
| Insulin§ (mU/L) | † | 7.5 (7.13 to 7.88) | 7.43 (6.89 to 8.02) | 8.69 (8.09 to 9.33) | 10.36 (9.21 to 11.65) | 0.01* | <0.0001* |
| | ‡ | 7.7 (7.36 to 8.07) | 7.68 (7.16 to 8.24) | 8.31 (7.77 to 8.88) | 9.29 (8.32 to 10.37) | 0.15 | 0.005* |
| **Inflammatory and endothelial markers (mean (95% CI))** | | | | | | | |
| CRP§ (mg/L) | † | 1.24 (1.13 to 1.36) | 1.36 (1.19 to 1.57) | 1.56 (1.37 to 1.78) | 1.67 (1.35 to 2.08) | 0.01* | 0.07 |
| | ‡ | 1.27 (1.16 to 1.39) | 1.4 (1.22 to 1.61) | 1.5 (1.32 to 1.71) | 1.51 (1.21 to 1.89) | 0.04* | 0.37 |
| IL-6§ (pg/mL) | † | 2.88 (2.73 to 3.05) | 3.01 (2.76 to 3.28) | 3.19 (2.95 to 3.45) | 3.58 (3.14 to 4.09) | 0.03* | 0.01* |
| | ‡ | 2.92 (2.76 to 3.08) | 3.06 (2.81 to 3.33) | 3.13 (2.89 to 3.39) | 3.34 (2.92 to 3.81) | 0.10 | 0.15 |
| vWF (IU/dL) | † | 132.65 (127.06 to 138.24) | 127.34 (118.81 to 135.87) | 134.92 (126.85 to 142.98) | 148.03 (135.05 to 161.01) | 1.00 | 0.02* |
| | ‡ | 132.76 (127.1 to 138.41) | 127.64 (119.03 to 136.24) | 134.88 (126.7 to 143.07) | 147.74 (134.39 to 161.1) | 0.96 | 0.03* |
| **Lung function (mean (95% CI))** | | | | | | | |
| FEV1 (L) | † | 2.52 (2.48 to 2.57) | 2.46 (2.39 to 2.52) | 2.37 (2.31 to 2.44) | 2.21 (2.11 to 2.31) | <0.0001 | <0.0001 |
| | ‡ | 2.52 (2.48 to 2.56) | 2.44 (2.38 to 2.51) | 2.38 (2.32 to 2.44) | 2.26 (2.15 to 2.36) | <0.0001 | <0.0001 |
| **Cardiac markers (mean (95% CI))** | | | | | | | |
| NT-pro-BNP§ (pg/mL) | † | 122.31 (110.53 to 135.34) | 110.95 (95.15 to 129.37) | 135.15 (117.08 to 156.02) | 171.15 (134.5 to 217.78) | 0.70 | 0.01 |
| | ‡ | 123.28 (111.4 to 136.43) | 111.2 (95.39 to 129.62) | 135.76 (117.52 to 156.82) | 167.33 (130.92 to 213.88) | 0.82 | 0.02 |
| cTnT§ (pg/mL) | † | 10.33 (9.83 to 10.85) | 10.45 (9.7 to 11.27) | 11.59 (10.8 to 12.43) | 12.79 (11.37 to 14.38) | 0.03 | 0.004 |
| | ‡ | 10.41 (9.91 to 10.93) | 10.52 (9.77 to 11.33) | 11.31 (10.55 to 12.13) | 12.3 (10.93 to 13.86) | 0.12 | 0.03 |

*denotes statistical significant

†Adjusted for age.

‡Adjusted for age, BMI, social class and physical activity.

§Geometeric mean.

¶Maximum n in group, varies slightly with missing covariate data.

CRP, C reactive protein; cTnT, cardiac troponin T; DBP, diastolic blood pressure; FEV1, forced expiratory volume in 1 s; HbA1c, glycated haemoglobin; HDL-C, high density lipoprotein cholesterol; IL-6, interleukin 6; NT-pro-BNP, N-terminal pro-brain natriuretic peptide; SBP, systolic blood pressure; vWF, von Willebrand factor.

(1.06 to 2.7)) and cTnT (OR (95% CI)=1.75 (1.09 to 2.8)) compared with those who reported no daytime sleep. However, this was attenuated following adjustment for prevalent diabetes, use of antihypertensive medication and depression (OR (95% CI)=1.59 (0.99 to 2.55)) and (OR (95% CI)=1.62 (1.00 to 2.62)), respectively, although it remained significant for cTnT. We further looked at men with an NT-pro-BNP of ≥400 pg/mL, the diagnostic threshold for HF. Men who reported EDS had a higher odds of having an NT-pro-BNP of ≥400 pg/mL (OR (95% CI)=1.88 (1.15 to 3.1)). This association remained following adjustment for prevalent diabetes, CKD, use of antihypertensive medication and depression (OR (95% CI)=1.72 (1.03 to 2.87)). Exclusion of men using anxiolytic and hypnotic medications did not significantly alter this (OR (95% CI)=1.76 (1.05 to 2.95)).

### Sleep patterns and non-invasive markers of arterial disease

Table 4 shows the associations between sleeping patterns and non-invasive vascular measurements. There was a significant association between those sleeping <6 hours and mean PWV and distensibility when compared with those sleeping 6–8.9 h in the fully adjusted model (p=0.04 and p=0.03, respectively). Men who slept <6 hours during the night-time were 1.6 times more likely to have a low (lowest quintile) mean distensibility compared with those who slept 6–8.9 hours in the fully adjusted model (OR (95% CI)=1.56 (1.09 to 2.23)). There was no significant association between daytime sleep and markers of arterial stiffness. There was no association between sleeping patterns and CIMT in the study population.

### DISCUSSION

We have confirmed previous reports of an association between night-time sleep duration[4] and daytime sleep duration[11 12] with metabolic risk factors. Our findings extend previous studies by investigating the association between EDS and a wide range of cardiac markers and vascular measures not previously examined. Our study has shown that in a representative sample of British men aged 71–92 years with no prior history of heart attack or HF, those who sleep more than 1 h during the day have a number of adverse characteristics including higher mean levels of cardiac markers, metabolic risk factors, inflammatory and endothelial markers and reduced lung function. These findings may point to pathways for the association observed between EDS and CVD and all-cause mortality.[8]

### EDS and metabolic and cardiac risk markers

EDS, defined in our study as daytime sleep of >1 h, is associated with many adverse characteristics such as obesity, physical inactivity, diabetes, breathlessness, use of antihypertensive therapy and with an adverse pattern of metabolic risk factors including HbA1c, glucose and insulin. Our results confirm previous cross-sectional analyses that have shown EDS to be associated with the metabolic syndrome and longer habitual napping

**Table 4** Sleeping patterns versus non-invasive vascular measurements

| | | CIMT (mm) | | PWV (m/s) | | Distensibility ($10^{-3}$ kPa$^{-1}$) | |
|---|---|---|---|---|---|---|---|
| | | Age adjusted mean (95% CI) | Additional adjustment*mean (95% CI) | Age adjusted mean (95% CI) | Additional adjustment mean (95% CI) | Age adjusted mean (95% CI) | Additional adjustment mean (95% CI) |
| Night-time sleep duration (hours) | <6 (n=209†) | | 0.81 (0.78 to 0.83) | 10.48 (10.24 to 10.71) | 10.45 (10.22 to 10.69) | 11.66 (11.11 to 12.21) | 11.62 (11.06 to 12.18) |
| | 6–8.9 (n=1125†) | | 0.81 (0.8 to 0.82) | 10.18 (10.08 to 10.27) | 10.18 (10.08 to 10.28) | 12.29 (12.06 to 12.53) | 12.31 (12.08 to 12.55) |
| | p | | 0.64 | 0.07 | 0.11 | 0.06 | 0.06 |
| Daytime sleep duration (hours) | 0 (n=677†) | | 0.81 (0.79 to 0.82) | 10.25 (... to ...38) | 10.25 (10.12 to 10.38) | 12.23 (11.93 to 12.53) | 12.2 (11.89 to 12.5) |
| | <1 (n=290†) | | 0.81 (0.79 to 0.82) | 10.15 (9.96 to 10.35) | 10.16 (9.96 to 10.36) | 12.49 (12.03 to 12.96) | 12.5 (12.04 to 12.97) |
| | 1 (n=334†) | | 0.8 (0.79 to 0.82) | 10.2 (10.01 to 10.38) | 10.2 (10.01 to 10.38) | 11.81 (11.38 to 12.25) | 11.88 (11.44 to 12.32) |
| | >1 (n=128†) | | 0.84 (0.81 to 0.87) | 10.29 (9.99 to 10.6) | 10.29 (9.98 to 10.6) | 11.96 (11.25 to 12.67) | 12.12 (11.39 to 12.85) |
| | p | | 0.17 | 0.80 | 0.85 | 0.18 | 0.30 |

\* Adjusted for age, BMI, social class and physical activity.
† Maximum n in group.
CIMT, carotid intima-media thickness; PWV, pulse wave velocity.

duration to be associated with an increased risk of impaired fasting glucose and diabetes mellitus.[11][12] There was a significant linear relationship between increasing daytime sleep and mean CRP levels, with those reporting EDS having higher levels of mean CRP. This confirms results from a previous study that showed self-reported daytime napping is associated with higher levels of CRP in older British adults.[26] The association seen between daytime sleep and these markers of inflammation may provide an explanation for the excess all-cause mortality observed in those with daytime napping or EDS.[8][9] While both of these studies have adjusted for diabetes, neither has adjusted for inflammation.

We found a significant association between EDS and high levels of NT-pro-BNP and cTnT, both of which are biomarkers of HF.[27][28] NT-pro-BNP is produced in response to cardiac wall stress and is a strong predictor of HF.[22] Troponin T is a plasma marker of myocyte necrosis, which has been positively associated with incident CVD, HF and all-cause mortality.[29] Most notably men who reported EDS had a higher odds of having NT-pro-BNP of $\geq 400\,pg/mL$ (the diagnostic threshold for HF). As far as we are aware, this is the first study to look at the association between EDS and these cardiac markers.

A strong association was also seen between EDS and elevated vWF (marker of endothelial dysfunction) and low FEV1, markers associated with HF. These findings suggest that EDS may be an early indicator of HF. This supports a recent report from our study when the men were aged 63–82 years which showed EDS to be associated with the risk of developing overt HF during 9-year follow-up.[7] In that analyses cardiac markers and lung function were not available.

The association between EDS and HF may be explained by underlying respiratory illness such as obstructive sleep apnoea (OSA). Indeed within our cohort, EDS was significantly associated with obesity, self-reported breathlessness and reduced lung function. EDS may be a result of disturbed night-time sleep and difficulty falling asleep, both of which have been associated with HF.[30][31] However, the associations we observed with cardiac markers and lung function were not seen in men with a short or long duration of night-time sleep and further analyses showed no association between these markers and those reporting difficulty falling asleep.

## Non-invasive markers of arterial disease

In contrast to night-time sleep, we did not observe any significant association between daytime sleep and markers of arterial stiffness. Short night-time sleep but not long night-time sleep was associated with arterial stiffness (PWV and carotid distensibility), when compared with those sleeping 6–8.9 hours. There was no association between sleeping patterns and CIMT in the study population. This is in contrast to previous studies that have shown an association between short night-time sleep[17] and EDS[5] with CIMT. The differences observed may be due to use of brachial-ankle PWV,[13][15] rather than carotid-femoral

PWV, different classification of long night-time sleep[13][14] and a younger study population.[15]

## Strengths and limitations

The strengths of our study include the fact that it is a study of older men, a group who are at high risk of vascular disease and the wide range of vascular parameters that we have looked into. Limitations of our study include the lack of representation of women, middle-aged populations and ethnic minority groups, which limits generalisability. Our data is based on self-reported sleeping patterns at a single time point, and it is possible that night-time and daytime sleep duration have been misclassified. Our analysis of night-time sleep relies on average duration of night-time sleep, and we do not have measures of interruption or total time spent in bed. As we do not have polysomnography data, we were unable to fully exclude OSA as a contributing factor to our findings. Data regarding occupation were not available, and we were unable to assess if previous shift work, which has been reported to be associated with cardiovascular and metabolic diseases, is related to the adverse cardiovascular profile we noted in those with EDS. We appreciate that our results are based on a cross-sectional analysis, and therefore causality cannot be established and the direction of the effect is unknown. It is also possible that there is an element of survival bias in our data; it may be that men who attended the re-examination were healthier, and this may account for the null associations seen.

## Conclusions

EDS, defined as daytime sleep of more than 1 hour, is associated with an adverse cardiovascular profile and may be an early indicator of HF. Clinically, this is important in the assessment of older patients who seek medical advice for EDS, in whom indicators of HF should be sought and further investigations carried out if necessary.

**Contributors** SGW initiated the concept and design of the paper. SZ analysed the data and drafted the manuscript. SGW, JPJH, EAE and PHW contributed to the interpretation of data. OP contributed to the analysis of the paper. EAE, JPJH, LL, PHW and SGW contributed to the acquisition of the data. All authors revised it critically for important intellectual content and approved the final version of the manuscript.

**Funding** The British Regional Heart Study is a British Heart Foundation (BHF) research group. This work was supported by a British Heart Foundation programme grant (RG/13/16/30528) and project grant (PG/09/024, PG/13/41/30304).

**Competing interests** None declared.

**Ethics approval** Ethical approval for data collection was obtained from the relevant research ethics committee.

**Provenance and peer review** Not commissioned; externally peer reviewed.

**Data sharing statement** Further details of the study can be found on http://www.ucl.ac.uk/pcph/research-groups-themes/brhs-pub. For general data sharing enquiries, please contact Lucy Lennon at l.lennon@ucl.ac.uk.

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
