## [Reviewer comments · BMJ Open]

ARTICLE DETAILS

TITLE (PROVISIONAL)	Self-reported sleep duration and napping, cardiac risk factors and markers of subclinical vascular disease: cross-sectional study in older men
AUTHORS	Zonoozi, Shahrzad; Ramsay, Sheena; Papacosta, Olia; Lennon, Lucy; Ellins, Elizabeth; Halcox, Julian; Whincup, Peter; Wannamethee, Goya

VERSION 1 - REVIEW

REVIEWER	Patricia Wong Department of Psychology, University of Pittsburgh, Pittsburgh, PA, USA
REVIEW RETURNED	02-Mar-2017

GENERAL COMMENTS	The present study is a very comprehensive examination of sleep (both daytime napping, an understudied factor, and nighttime sleep) and various metabolic and cardiovascular risk factors. The following are some questions and comments regarding the manuscript: • What is the correlation between nighttime and daytime sleep? Are participants who slept more during the daytime “making up” for less nighttime sleep? If so, are the effects of sleep duration and napping independent of each other or actually driving one another. In other words, can the authors report the correlation and add in discussion as to whether they have controlled for nighttime sleep when examining napping and vice versa?• Please clarify this sentence, it’s not clear if the authors meant self-report, a doctor made the diagnosis, or both: “The presence of comorbidity was based on the men self-reporting a doctor made diagnosis and those with a history of heart attack or HF were excluded from the analysis (n = 251).” Pg5
--

	 • Do the authors have data as to whether any of the subjects worked rotating or night-shift work? There is growing evidence showing that such shift work associates with metabolic and cardiovascular risk factors, and predicts to greater risk for various illnesses (e.g., cardiovascular disease, diabetes). If possible, the authors should examine any such relationship in their dataset or acknowledge it as a discussion point if that data is not available. • The tables are well organized. The authors condensed a lot of data and analyses into an understandable and approachable format. • The discussion and statements about limitations/strengths are appropriate.
--	---

REVIEWER	Empana, Jean-Philippe INSERM; France
REVIEW RETURNED	17-Mar-2017

GENERAL COMMENTS	In this cross sectional analysis of the well-known BHRS study, the authors investigate associations of night time sleep and daytime sleep with cardiovascular and metabolic risk factors, structural and functional arterial parameters, and a panel of blood biomarkers, in older men aged 72-90 years. Novelty concerns the investigation of daytime sleepiness with structural and functional arterial parameters, and a panel of blood biomarkers including IL6, vWF and BNP. The findings suggest that daytime sleepiness might be associated with early sign of heart failure, which has strong clinical implications. The authors acknowledge adequately their main limitations, including the selected nature of their population (older men only, possible survival bias) the lack of data on OSA, and the cross sectional study design of their study. However, with the data at hand, the authors were able to provide clear and relevant study results that may have clinical implications. I have only minor comments  1. The limit of quantification of cardiac Troponin together with the % of troponin detected should be given 2. Is there any rationale or prior evidence for defining daytime sleepiness as a daytime sleep >1h ? 3. What about sleep medications: any data ? what if adjusting on that ? 4. Any missing data and any impact on study results ?
---

	5. As a cross sectional study, some associations could be bi directional. For instance, abnormalities in arterial stiffness or an underlying heart failure might "predict" daytime sleep. Hence, the authors should look whether PWV or NT-proBNP>400 pg/ml, considered here as the exposures are associated with daytime sleep (outcome).
--	--

VERSION 1 – AUTHOR RESPONSE

Reviewer 1

1. The limit of quantification of cardiac Troponin together with the % of troponin detected should be given.

The lowest detectable value of cTnT was 5 and for anyone with a recorded reading of < 5, we used half of the lowest detectable value (i.e. 2.5). This has now been included on page 5.

2. Is there any rationale or prior evidence for defining daytime sleepiness as a daytime sleep >1h? As per a previous report from this study, men who reported daytime sleep of >1 hour were at higher risk of incident heart failure and therefore this cut off was also used in our paper (Wannamethee SG, Papacosta O, Lennon L, Whincup PH. Self-Reported Sleep Duration, Napping, and Incident Heart Failure: Prospective Associations in the British Regional Heart Study. Journal of the American Geriatrics Society. 2016.).

3. What about sleep medications: any data? What if adjusting on that?

Information regarding sleep medications is available and has now been included. 31 men reported using hypnotic or anxiolytic medications. There was a significant association between night-time sleep and use of hypnotic or anxiolytic medications. There was no significant association between use of these medications and daytime sleep. Even after exclusion of men using anxiolytic and hypnotic medications, those reporting EDS had a higher odds of having an NT-pro BNP of ≥ 400 pg/ml. This information has now been included on pages 8 and 10.

4. Any missing data and any impact on study results?

Of the 1471 men included in the study, 1436 (97.6%) completed the questions on night-time and daytime sleep duration (35 missing). This is reported on page 6 of the article. All of those who responded to the questions on sleeping patterns had at least one blood measurement. Given the low number of missing values, it was not felt this would have any significant impact on the results. We have discussed the potential for survival bias in the strengths and limitations section of the article (page 13).

5. As a cross sectional study, some associations could be bi directional. For instance, abnormalities in arterial stiffness or an underlying heart failure might "predict" daytime sleep. Hence, the authors should look whether PWV or NT-proBNP>400 pg/ml, considered here as the exposures are associated with daytime sleep (outcome).

We did not find an association between EDS and PWV. We accept that the direction of the association between daytime sleep and NT-proBNP cannot be established and have clarified this in the text on page 13.

Reviewer 2

1. What is the correlation between nighttime and daytime sleep? Are participants who slept more during the daytime "making up" for less nighttime sleep? If so, are the effects of sleep duration and napping independent of each other or actually driving one another. In other words, can the authors

report the correlation and add in discussion as to whether they have controlled for nighttime sleep when examining napping and vice versa?

As reported in the results section, “Night-time sleep duration was significantly associated with EDS ($p = 0.004$), with a smaller proportion of those sleeping ≥ 9 hours reporting EDS. Daytime sleep duration was significantly associated with night-time sleep of less than 6 hours ($p = 0.02$).”

The main results of our study show that daytime sleep is significantly associated with NT-pro BNP and troponin. No association was seen between night-time sleep and these blood markers and therefore no further adjustment was carried out. Both night-time and daytime sleep are associated with metabolic risk factors and mean levels of glucose and insulin remained significantly higher in those with EDS ($p = 0.03$ and 0.01 respectively) even after further adjustment for prevalent diabetes and night-time sleep. This has now been included in the article on page 9. Furthermore, even following adjustment for daytime sleep, night-time sleep remained significantly associated with HbA1c – this has also been included on page 8.

2. Please clarify this sentence, it’s not clear if the authors meant self-report, a doctor made the diagnosis, or both: “The presence of comorbidity was based on the men self-reporting a doctor made diagnosis and those with a history of heart attack or HF were excluded from the analysis ($n = 251$).”
Pg5

This has been changed to: “The men were asked if a doctor had ever told them they had a heart attack or HF and these men were excluded from the analysis ($n = 251$).”

3. Do the authors have data as to whether any of the subjects worked rotating or night-shift work? There is growing evidence showing that such shift work associates with metabolic and cardiovascular risk factors, and predicts to greater risk for various illnesses (e.g., cardiovascular disease, diabetes). If possible, the authors should examine any such relationship in their dataset or acknowledge it as a discussion point if that data is not available.

Unfortunately this data is not available for our cohort and this has now been added to the discussion.

VERSION 2 – REVIEW

REVIEWER	Patricia Wong University of Pittsburgh, Pittsburgh, PA, USA
REVIEW RETURNED	28-Apr-2017

GENERAL COMMENTS	It is recommended the authors include the correlation statistic (not just the p-value) for the correlation between napping and nighttime sleep to show whether the factors have a small, moderate or large correlation. The interest in daytime napping has been growing, and so descriptive information such as this is helpful to compare between populations and studies. Otherwise, the authors have comprehensively addressed all reviewer questions and comments.
--

REVIEWER	Empana, Jean-Philippe INSERM U970, Paris, France
REVIEW RETURNED	01-May-2017

GENERAL COMMENTS	The authors have addressed adequately my concerns and showed that their study results were robust. I appreciate the self-evaluation of the data at hands and inherent limitations. This paper opens some avenues regarding the mechanisms underlying EDS and CVD, at least in older adult men.
--

VERSION 2 – AUTHOR RESPONSE

Many thanks for your response. Further to the comments from the reviewers, the manuscript has been changed to include the correlation statistic (not just the p-value) for the correlation between daytime and night-time sleep:

"Night-time sleep duration was significantly associated with EDS ($p = 0.02$; correlation coefficient $r = -0.063$), with a smaller proportion of those sleeping ≥ 9 hours reporting EDS."